# VARIATIONAL NEURAL CELLULAR AUTOMATA

**Rasmus Berg Palm**[1]**, Miguel González-Duque**[2]**, Shyam Sudhakaran**[3] **& Sebastian Risi**[4]
Creative AI Lab
IT University of Copenhagen
Copenhagen, Denmark
{[1]`rasmb`,[2]`migd`,[4]`sebr`}`@itu.dk`, [3]`shyamsnair@protonmail.com`

## ABSTRACT

In nature, the process of cellular growth and differentiation has lead to an amazing diversity of organisms — algae, starfish, giant sequoia, tardigrades, and orcas are all created by the same generative process. Inspired by the incredible diversity of this biological generative process, we propose a generative model, the Variational Neural Cellular Automata (VNCA), which is loosely inspired by the biological processes of cellular growth and differentiation. Unlike previous related works, the VNCA is a proper probabilistic generative model, and we evaluate it according to best practices. We find that the VNCA learns to reconstruct samples well and that despite its relatively few parameters and simple local-only communication, the VNCA can learn to generate a large variety of output from information encoded in a common vector format. While there is a significant gap to the current state-of-the-art in terms of generative modeling performance, we show that the VNCA can learn a purely self-organizing generative process of data. Additionally, we show that the VNCA can learn a distribution of stable attractors that can recover from significant damage.

## 1 INTRODUCTION

The process of cellular growth and differentiation is capable of creating an astonishing breadth of form and function, filling every conceivable niche in our ecosystem. Organisms range from complex multi-cellular organisms like trees, birds, and humans to tiny microorganisms living near hydrothermal vents in hot, toxic water at immense pressure. All these incredibly diverse organisms are the result of the same generative process of cellular growth and differentiation.

Cellular Automata (CA) are computational systems inspired by this process of cellular growth and differentiation, where "cells" iteratively update their state based on the state of their neighbor cells

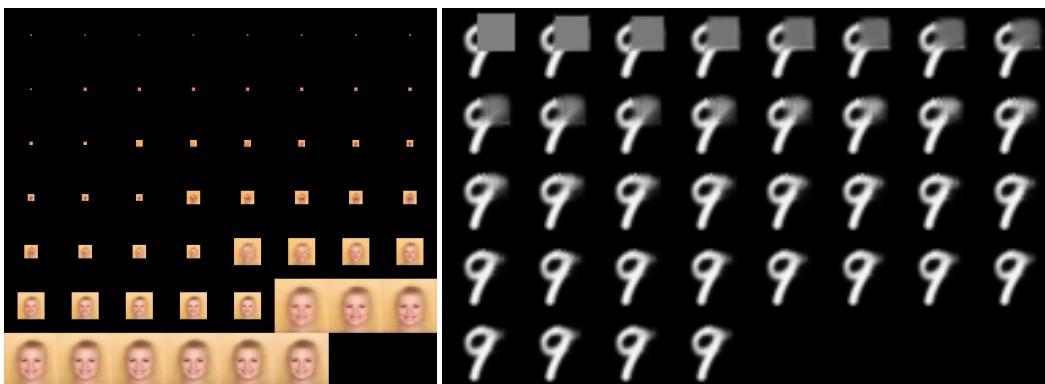

Figure 1: **VNCA self-organized growth and damage recovery.** Left: The VNCA has learned a self-organising generative process that generates faces, from an initial random seed of cells from $\mathcal{N}(0, I)$. Time goes top-down and left-to-right. Every eight steps all the cells duplicate, which can be seen on the diagonal. Right: A damage recovery sequence for an MNIST digit.

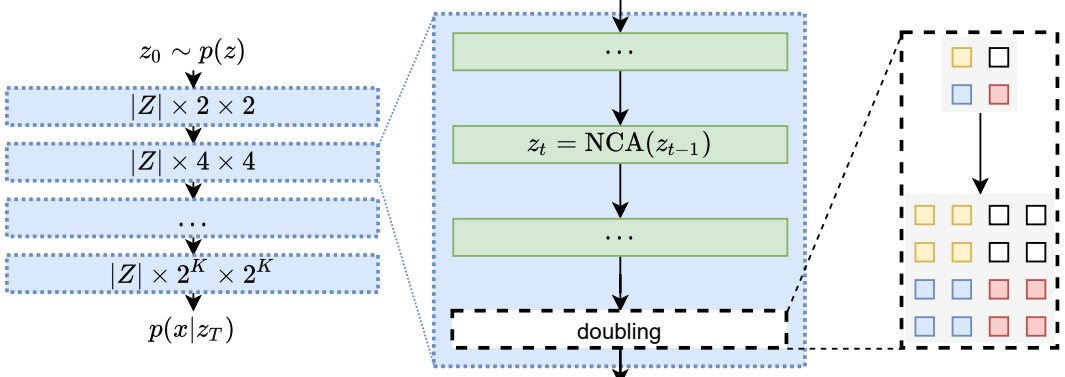

Figure 2: VNCA overview. Left: Generative process. $z_0$ is sampled from $p(z)$ and acts as an initial $2 \times 2$ seed of cells, which grows through a series of K doubling NCA steps. $|Z|$ denotes the size of the latent space, which is identical to the size of the cell states. Finally, the last cell hidden state conditions the parameters of the likelihood distribution $p(x|z)$. Middle: Each doubling NCA step consists of a number of NCA steps followed by a doubling operation. At each NCA step, the cells can only communicate with their immediate neighbors. Right: The doubling operator doubles the cell grid as depicted, where the color indicates the state vector of each cell.

and the rules of the CA. Even with simple rules, CA can exhibit complex and surprising behavior — with just four simple rules, Conway's Game of Life is Turing complete. Neural Cellular Automata (NCA) are CA where the cell states are vectors, and the rules of the CA are parameterized and learned by neural networks (Mordvintsev et al., 2020; Nichele et al., 2017; Stanley & Miikkulainen, 2003; Wulff & Hertz, 1992). NCAs have been shown to learn to generate images, 3D structures, and even functional artifacts that are capable of regenerating when damaged (Mordvintsev et al., 2020; Sudhakaran et al., 2021). While these results are impressive, the NCAs can only generate (and regenerate) the single artifact it is trained on, lacking the diverse generative properties of current probabilistic generative models. In this paper, we introduce the VNCA, an NCA based architecture that addresses these limitation while only relying on local communication and self-organization.

Generative modeling is a fundamental task in machine learning, and several classes of methods have been proposed. Probabilistic generative models are an attractive class of models that directly define a distribution over the data $p(x)$, which enable sampling and computing (at least approximately) likelihoods of observed data. The average likelihood on test data can then be compared between models, which enables fair comparison between generative models. The Variational Auto-Encoder (VAE) is a seminal probabilistic generative model, which models the data using a latent variable model, such that $p(x) = \int_z p_\theta(x|z)p(z)$, where $p(z)$ is a fixed prior and $p_\theta(x|z)$ is a learned decoder (Kingma & Welling, 2013; Rezende et al., 2014). The parameters of the model is learned by maximizing the Evidence Lower Bound (ELBO), a lower bound on $\log p(x)$, which can be efficiently computed using amortized variational inference (Kingma & Welling, 2013; Rezende et al., 2014).

On a high level, our proposed generative model combines NCAs with VAEs, by using a NCA as the decoder in a VAE. We perform experiments with a standard NCA decoder and a novel NCA architecture, which duplicates all cells every $M$ steps, inspired by cell mitosis. This results in better generative modeling performance and has the computational benefit that the VNCA only computes cell updates for the living cells at each step, which are relatively few early in growth. While our model is inspired by the biological process of cellular growth and differentiation, it is not intended to be an accurate model of this process and is naturally constrained by what we can efficiently compute.

Other works have explored similar directions. The VAE-NCA proposed in Chen & Wang (2020), is actually, despite the name, not a VAE since it does not define a generative model or use variational inference. Rather it is an auto-encoder that decodes a set of weights, which parameterize a NCA that reconstructs images. NCAM (Ruiz et al., 2020) similarly uses an auto-encoder to decode weights for a NCA, which then generates images. In contrast to both of these, our method *is* a generative

model, uses a single learned NCA, and samples the initial state of the seed cells from a prior $p(z)$. StampCA (Frans, 2021) also learns an auto-encoder but similarly encodes the initial hidden NCA state of a central cell. None of these are generative models, in the sense that it's not possible to sample data from them, nor do they assign likelihoods to samples. While previous works all show impressive reconstructions, it is impossible to evaluate how good generative models they really are. In contrast, we aim to offer a fair and honest evaluation of the VNCA as a generative model.

Graph Neural Networks (GNNs) can be seen as a generalization of the local-only communication in NCA, where the grid neighborhood is relaxed into an arbitrary graph neighborhood (Grattarola et al., 2021). GNNs compute vector-valued messages along the edges of the graph that are accumulated at the nodes, using a shared function among all nodes and edges. GNNs have been successfully used to model molecular properties (Gilmer et al., 2017), reasoning about objects and their interactions (Santoro et al., 2017), and learning to control self-assembling morphologies (Pathak et al., 2019).

Powerful generative image models have been proposed based on the variational auto-encoder framework. A common distinction is whether the decoder is auto-regressive or not, i.e., whether the decoder conditions its outputs on previous outputs, usually those above and to the left of the current pixel (Van den Oord et al., 2016; Salimans et al., 2017). Auto-regressive decoders are so effective at modelling $p(x)$ directly that they have the tendency to ignore the latent codes thus failing to reconstruct images well (Alemi et al., 2018). Additionally, they are expensive to sample from since sampling must be done one pixel at a time. Our decoder is not auto-regressive, thus cheaper to sample from, and more similar to e.g., BIVA (Maaløe et al., 2019) or NVAE (Vahdat & Kautz, 2020). Contrary to BIVA, NVAE and other similar deep convolution-based VAEs, our VNCA defines a relatively simple decoder function that only consider the immediate neighborhood of a cell (pixel) at each step and is applied iteratively over a number of steps. The VNCA thus aims to learns a self-organising generative process.

NCAs and self-organizing systems in general, represent an exciting research direction. The repeated application of a simple function that only relies on local communication lends itself well to a distributed leaderless computation paradigm, useful in, for instance, swarm robotics (Rubenstein et al., 2012). Additionally, such self-organizing systems are often inherently robust to perturbations (Mordvintsev et al., 2020; Najarro & Risi, 2020; Tang & Ha, 2021; Sudhakaran et al., 2021). In the VNCA, we show that this robustness allows us to set a large fraction of the cells' states to zero and recover almost perfectly by additional iterations of growth (Fig. 1, right). By introducing an NCA that is a proper generative probabilistic model, we hope to further spur the development of methods at the intersection of artificial life and generative models.

## 2 VARIATIONAL NEURAL CELLULAR AUTOMATA

The VNCA defines a generative latent variable model with $z \sim p(z)$ and $x \sim p_\theta(x|z)$, where $p(z) = \mathcal{N}(z|\mu = 0, \Sigma = I)$ is a Gaussian with zero mean and diagonal co-variance and $p_\theta(x|z)$ is a learned decoder, with parameters $\theta$. The model is trained by maximizing the evidence lower bound (ELBO),

$$\log p(x) - D_{\text{KL}}(q_\phi(z|x) \parallel p(z|x)) = -D_{\text{KL}}(q_\phi(z|x) \parallel p(z)) + \mathbb{E}_{z \sim q_\phi(z|x)} \log p_\theta(x|z),$$

where, $q_\phi(z|x)$ is a learned encoder, with parameters $\phi$, which learns to perform amortized variational inference. Since $D_{\text{KL}} \geq 0$, the right hand side is a lower bound on $\log p(x)$ and increasing it either increases $\log p(x)$ or decreases $D_{\text{KL}}(q_\phi(z|x) \parallel p(z|x))$, which is how closely the amortized variational inference matches the true unknown posterior $p(z|x)$. By using the reparameterization trick, the sampling, $z \sim q_\phi(z|x)$, is differentiable, and both $\phi$ and $\theta$ can be learned efficiently with stochastic gradient ascent (Kingma & Welling, 2013). The $D_{\text{KL}}$ term on the right hand side can be weighted with a parameter $\beta$ to control whether the VAE should prioritize better reconstructions ($\beta < 1$), or better samples ($\beta > 1$) (Higgins et al., 2016). For $\beta > 1$ this is still a (looser) lower bound on $\log p(x)$.

The decoder of the generative process, $p_\theta(x|z)$, is based on a NCA (Fig. 2). The NCA defines a recurrent computation over vector-valued "cells" in a grid. At step $t$ of the NCA, the cells compute an additive update to their state, based only on the state of their neighbor cells in a $3 \times 3$ neighborhood,

$$z_{i,t} = z_{i,t-1} + u_\theta(\{z_{j,t-1}\}_{j \in N(i)}), \tag{1}$$

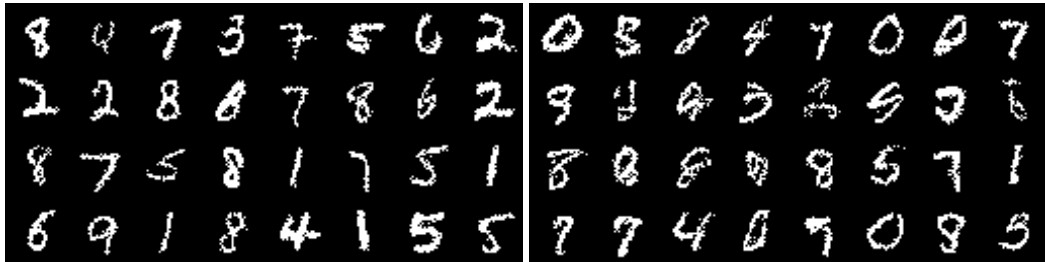

Figure 3: Left: Test set reconstructions. Right: Unconditional samples from the prior. The VNCA achieves $\log p(x) \geq -84.23$ nats evaluated with 128 importance weighted samples.

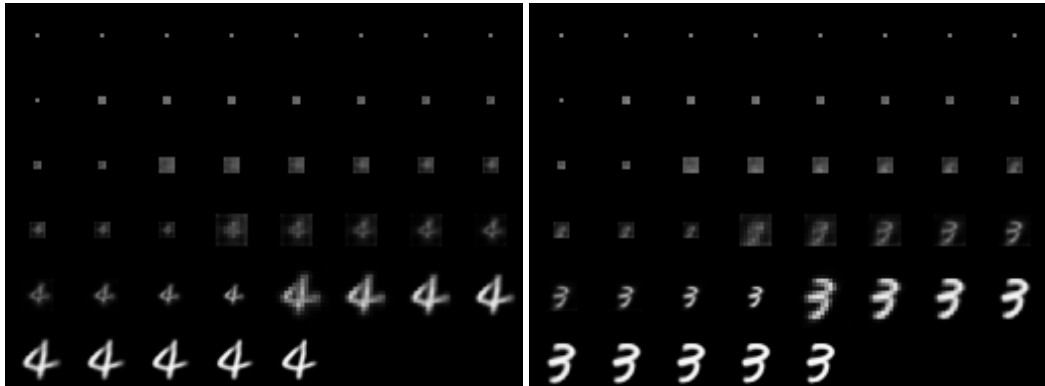

Figure 4: Self-organising MNIST, with the seed cell states sampled from the prior. Each $32 \times 32$ sub-image shows the digit after 1 step of NCA or doubling. Time flows top-to-bottom and left-to-right. The doubling steps can be seen on the diagonal. Note: this shows sample averages for clarity.

where $u_\theta$ is a learned update function, index $i$ denote a single cell and $N(i)$ denotes the indexes of the $3 \times 3$ neighborhood of cell $i$, including $i$. All cells are updated in parallel. In practice, $u_\theta$ is efficiently implemented using a Convolutional Neural Network (CNN) (Gilpin, 2019) with a single $3 \times 3$ filter followed by a number of $1 \times 1$ filters and non-linearities. The initial grid $z_0$ consists of a single sample from the latent space repeated in a $2 \times 2$ grid.

A traditional NCA defines whether a cell is alive or dead based on the cell state and its neighbors' cell states (Mordvintsev et al., 2020). This allows the NCA to "grow" from a small initial seed of alive cells. Our NCA uses a different approach inspired by cell mitosis. Every $M$ steps of regular NCA updates all the cells duplicate by creating new cells initialized to their own current state to the right and below themselves, "pushing" the existing cells out of the way as they do so. This is efficiently implemented using a "nearest" type image rescaling operation (Fig. 2, right). After $K$ doubling and $M(K + 1)$ steps of NCA updates, the final cell states $z_T$ condition the parameters of the likelihood distribution. For RGB images, we use a discrete logistic likelihood with $L$ mixtures (Salimans et al., 2017), such that the $10L$ first channels are the parameters of the discrete logistic mixture[1] and for binary images a Bernoulli likelihood with the first channel of each cell being $\log p$.

The update function is a single $3 \times 3$ convolution, followed by four residual blocks (He et al., 2016). Each block consists of a $1 \times 1$ convolution, Exponential Linear Unit (ELU) non-linearity (Clevert et al., 2015) and another $1 \times 1$ convolution. Finally, a single $1 \times 1$ convolution maps to the cell state size. This last convolution layers' weights and biases are initialized to zero to prevent the recurrent computation from overflowing. The number of channels for all convolutional layers, the size of the latent space, and the cell state are all 256. The total number of parameters in the NCA decoder is approximately 1.2M.

Since we are primarily interested in examining the properties of the NCA decoder, the encoder $q_\phi(z|x)$ is a fairly standard deep convolutional network. It consists of a single layer of $5 \times 5$ con-

---

[1]We use the implementation from `https://github.com/vlievin/biva-pytorch`

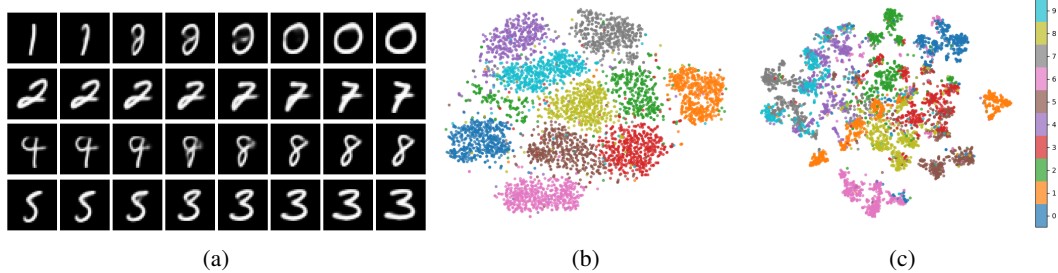

| (a) | (b) | (c) |

Figure 5: Exploring the latent space of VNCA. Fig. 5a shows several linear interpolations between random digits in the latent space of a VNCA trained on binarized MNIST. Figs. 5b and 5c show the result of reducing the dimensionality of 5000 digits chosen at random for the VNCA and for the deep convolutional baseline respectively. Notice how the latent space of VNCA has more t-SNE structure and cleaner separation of digit encodings. Note: this shows sample averages for clarity.

volution with 32 channels followed by four layers of $5 \times 5$ convolution with stride 2, where each of these layers has twice the number of channels as the previous layer. Finally, the down-scaled image is flattened and passed through a dense layer which maps to twice the latent space dimensionality. The first half encodes the mean and the second half the log-variance of a Gaussian. All convolutional layers are followed by ELU non-linearities. For a detailed model definition, see the appendix.

## 3 EXPERIMENTS

Except where otherwise noted, we use a batch size of 32, Adam optimizer (Kingma & Ba, 2014), $10^{-4}$ learning rate, $L = 1$ logistic mixture component, clip the gradient norm to 10 (Pascanu et al., 2013) and train the VNCA for 100.000 gradient updates. Architectures and hyper-parameters were heuristically explored and chosen based on their performance on the validation sets and the memory limits of our GPUs. We use a single sample to compute the ELBO when training and measure final log-likelihoods on the test set using 128 importance weighted samples, which gives a tighter lower bound than the ELBO (Burda et al., 2015). We report log likelihoods in nats summed over all the dimensions for MNIST and bits per dimension (bpd) for CelebA per convention for these datasets. The conversion between the two is bpd $= -\text{nats}/(HWC \ln 2)$, where $HWC$ is height, width and channels respectively. When visualizing the output of the generative models we show $x \sim p(x|z)$, i.e. samples from the generative model or the average, when the noise of the individual samples detracts from the analysis. When we show averages we note this in the figure captions.

### 3.1 MNIST

For our first experiment we chose MNIST, since it is widely used in the generative modeling literature and a relatively easy dataset. Since the VNCA generates images that have height and width that are powers of 2, we pad the MNIST images to $32 \times 32$ with zeros and use $K = 4$ doublings. We use the statically binarized MNIST from Larochelle & Murray (2011).

The VNCA learns a generative model of MNIST digits and also is capable of accurately reconstructing digits given a latent code by the encoder (Fig. 3). It achieves $\log p(x) \geq -84.23$ nats evaluated with 128 importance weighted samples. While not state-of-the-art, which is around $-77$ nats (Sinha & Dieng, 2021; Maaløe et al., 2019), it is a decent generative model. In comparison, the complex auto-regressive PixelCNN decoder achieves $-81.3$ (Van den Oord et al., 2016).

Fig. 4 shows the self-organization and growth of the digits. Initially, they are just uniform squares, but after doubling twice to a size of $8 \times 8$, it appears that the process is iterating towards a pattern. After doubling, the NCA steps shows signs of "enhancing" the image, moving towards a smoother, more natural-looking digit. The VNCA has learned a self-organizing process that converges to diverse naturally looking hand-written digits.

**Latent space analysis** We visualize in Fig. 5b the latent space of our VNCA by selecting 5000 random images from the binarized MNIST dataset, encoding them, and reducing their dimensional-

ity using t-Stochastic Neighbour Embedding (t-SNE) (van der Maaten & Hinton, 2008). We find that our dimensionally-reduced latent space cleanly separates digits, especially when compared to a deep convolutional baseline in Fig. 5c. This baseline is composed of the same encoder as the VNCA, followed by a symmetric deconvolutional decoder (see the appendix for details). Further, Fig. 5a shows four interpolations between the encodings of digits selected at random from the dataset.

## 3.2 CELEBA

The CelebA dataset contains 202,599 images of celebrity faces (Liu et al., 2015) and has been extensively used in the generative modeling literature. It is a significantly harder dataset to model than MNIST and allows us to fairly compare the VNCA against state-of-the-art generative methods. We use the version with images rescaled to $64 \times 64$.

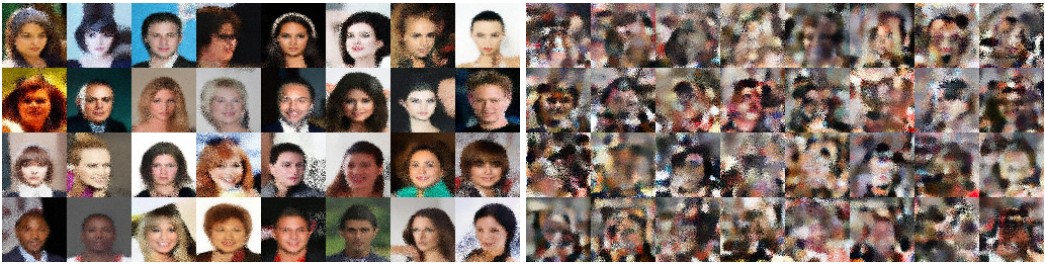

Figure 6: CelebA results. Left: Test set reconstructions. Right: Unconditional samples from the prior. The VNCA achieves $4.42$ bits per dimension on the test set evaluated with $128$ importance weighted samples.

The VNCA learns to reconstruct the images well, but the unconditional samples do not look like faces, although there are some faint face-like structures in some of the samples (Fig. 6). The VNCA achieves 4.42 bits per dimension evaluated with 128 importance weighted samples. This is far from state-of-the-art which is around 2 bits per dimension, using diffusion based models and deep hierarchical VAEs (Kim et al., 2021; Vahdat & Kautz, 2020).

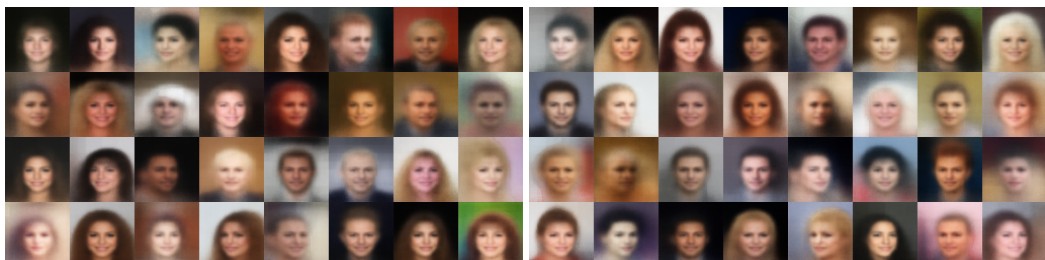

Figure 7: VNCA CelebA results when trained with $\beta = 100$. Left: Test set reconstructions. Right: Unconditional samples from the prior. Note: this shows sample averages for clarity. This version achieves much worse bits per dimension $\geq 5.40$, but visually the samples are much better. Note: this shows sample averages for clarity.

Seeing that the reconstructions are good, but the samples are poor, we train a VNCA with $\beta = 100$, which strongly encourages the VNCA to produce better samples at the cost of the reconstructions. See fig. 7 for the results. This VNCA is technically a much worse generative model with bits per dimension $\geq 5.40$, but visually the samples are much better. Individually the samples are noisy due to the large variance of the logistic distribution, so we visualize the average sample. For individual samples from the likelihood see the appendix.

With the improved visual samples, we can visualize the growth (Fig. 8). Here we see that the VNCA has learned a self-organizing generative process that generates faces from random initial seeds. Similar to the MNIST results, the images go through alternating steps of doubling and NCA steps that refine the pixellated images after a doubling.

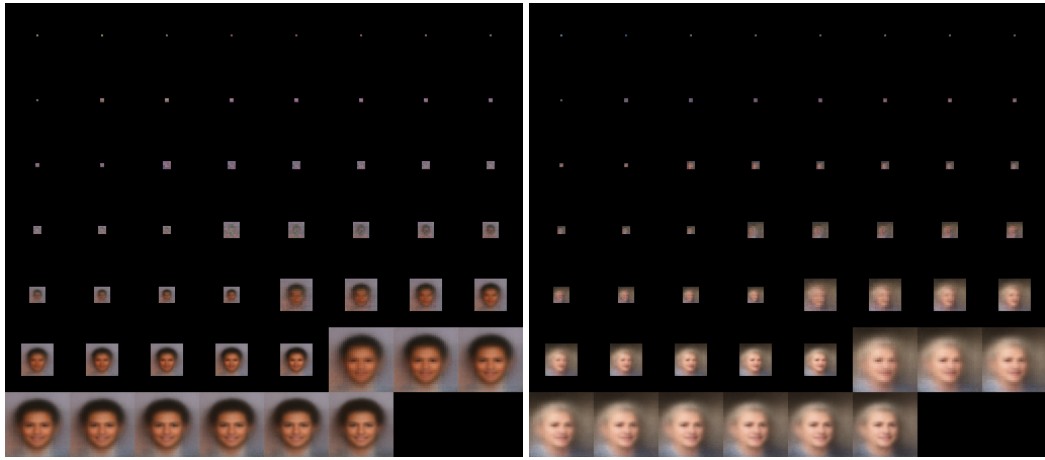

Figure 8: Growth of CelebA samples from the prior from a VNCA trained with $\beta = 100$. Note: this shows sample averages for clarity.

**Resilience to damage during growth**  Due to the iterative self-organising nature of the computation, the original NCA architectures were shown to be resilient in the face of damage, especially if trained for it (Mordvintsev et al., 2020). To examine the VNCAs inherent resilience to damage during growth, we perform an experiment in which we set the right half of all cell states to zero, at varying stages of growth. We compared the VNCA resilience to damage against an baseline using an identical encoder and a deep convolutional VAE with 5 up-convolutions as the decoder. To our surprise the baseline VAE decoder recover approximately similarly well from this damage, at similar stages in the decoding process. See Fig. 11 in the appendix for the results.

### 3.3 Optimizing for resilience to damage

Inspired by the damage resilience shown in (Mordvintsev et al., 2020) we investigate how resilient we can make the VNCA to damage by explicitly optimizing for it. In order to recover from damage we need to grow until the damage is recovered. However, due to the doubling layers, this would quickly result in an image larger than the original data. As such we modify the VNCA architecture by removing the doubling layers. Since this non-doubling VNCA variant does not changes the resolution we can run it for any number of steps. The NCA decoder of this variant is almost identical to the original NCA architecture proposed in (Mordvintsev et al., 2020). The only differences are that (1) we consider all cells alive at all time, since we found that the aliveness concept proposed in Mordvintsev et al. (2020) made training unstable, (2) we seed the entire $H \times W$ with $z_0$ instead of just the center and (3) we update all cells all the time. The only difference to the doubling VNCA is that we remove the doubling layers, and seed the entire grid with $z_0$, so eq. 1 still describes the update rule. See the appendix for a figure of this non-doubling variant.

**Pool and damage recovery training.**  In order to optimize for stability and damage recovery we adapt the pool and damage recovery training procedure proposed in Mordvintsev et al. (2020) to our generative modelling setting. During training half of every training batch is sampled from a pool of previous samples and their corresponding cell states after running the NCA, $z_T$. Half of these pool states $z_T$ are subjected to damage by setting the cell state to zero in a random $H/2 \times W/2$ square. After the NCA has run for $T$ steps, we put all the new $z_T$ states in the pool, shuffle the pool, and retain the first $P$ samples, where $P$ corresponds to the fixed size of our pool. We use $P = 1024$ for our experiments. See appendix A.3 for pseudo-code. The effect is that half of every training batch is normal, and half corresponds to running the NCA for an additional $T$ steps on a previous sample, of which half are damaged. For the half that corresponds to running the NCA for an additional $T$ steps, the gradient computation is truncated, corresponding to truncated backpropagation through time. As such, we are no longer maximizing the ELBO exactly, but in practice it still seems to work.

**MNIST.** We train a non-doubling VNCA with pool and damage training on MNIST. It uses a very simple update network consisting of a single $3 \times 3$ convolution, ELU non-linearity, and then a single $1 \times 1$ convolution. For each batch we sample $T$ uniformly from 32 to 64 steps. It is trained with a batch and latent size of 128. It attains worse log-likelihoods than the doubling VNCA with $\log p(x) \geq -90.97$ nats, however, the VNCA learns a distribution of attractors stable over hundreds of steps which recover almost perfectly from damage (Fig. 9). While this looks like in-painting or

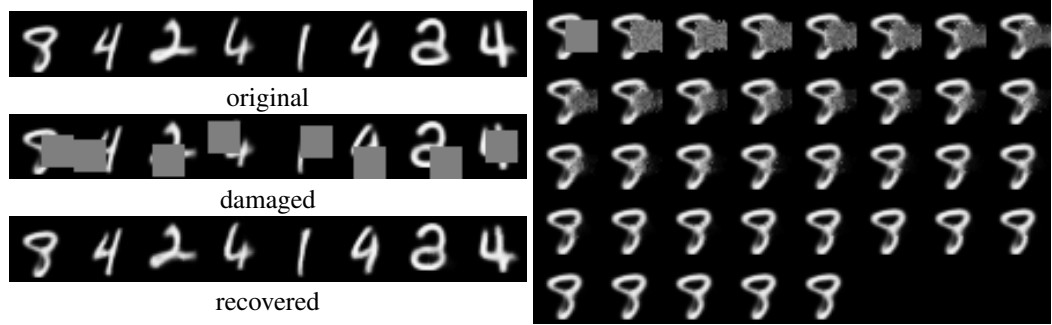

Figure 9: Damage recovery on MNIST samples. Left: From top to bottom: samples from the model, samples with damage applied, recovered samples after $T$ NCA steps. Right: details of the damage recovery process showing $T = 36$ steps of NCA growth on the leftmost damaged sample ($T$ is sampled uniformly from 32 to 64). Note: this shows sample averages for clarity.

data-imputation, it is not, and we should be careful not to compare to such methods. When doing in-painting the method only has access to the pixels of the damaged image, whereas here, each pixel corresponds to a vector valued cell state, that can potentially encode the contents of the entire image. Indeed this seems to be the case, since the VNCA almost perfectly recover the original input.

**CelebA.** We also train a non-doubling VNCA with pool and damage recovery on CelebA. The hyper-parameters are identical to the doubling variant except the data is rescaled to $32 \times 32$, we run it for 64 to 128 steps sampled uniformly, and use 128 hidden NCA states instead of 256, due to memory limitations. Similarly to MNIST, the non-doubling CelebA variant is a worse generative model, achieving 5.03 bits per dimension, on the easier $32 \times 32$ version of CelebA, but learns to recover from damage and learns attractors that are stable for hundreds of NCA steps (Fig. 10).

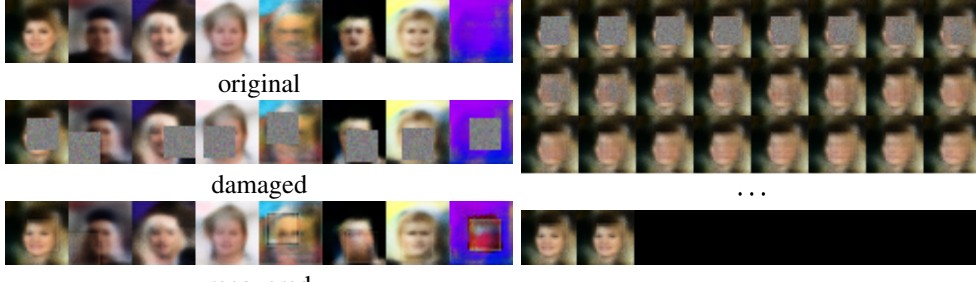

Figure 10: CelebA Damage Recovery. Left: From top to bottom: samples from the model, samples with damage applied, recovered samples after $T$ NCA steps. Right: details of the damage recovery process, cropped for brevity, showing the first 24 and the last 2 steps on the leftmost damaged sample. For all 121 steps, see the appendix. Note: this shows sample averages for clarity.

## 4 DISCUSSION

Inspired by the biological generative process of cellular division and differentiation, we proposed the VNCA, a generative model based on a self-organising & iterative computation, relying only on local cellular communication. We showed that, while far from state-of-the-art, it is capable of learning a

self-organising generative model of data. Further, we showed that it can learn a distribution of stable attractors of data that can recover from damage.

We proposed two variants of the VNCA. One based on doubling of cells at fixed intervals, inspired by cell mitosis, and a non-doubling variant. The non-doubling variant is better suited for damage recovery and long-term stability but it is a worse generative model. We hypothesize that (1) the doubling operation is a good inductive bias on image data, and that (2) it allows the cells to agree on a global structure early on. The communication in the early growth steps effectively enables the doubling VNCA to communicate over, what becomes, large distances, while the non-doubling NCA needs many steps to achieve something similar. The doubling VNCA also has the benefit that it is computationally cheaper since it is only in the last stage of growth that the computations are performed on the entire image. Compared to the non-doubling VNCA this results in a large speedup and lowers the memory requirements dramatically. How to combine the damage resilience of the non-doubling variant with the better generative modelling of the mitosis inspired variant is an exciting avenue for future research.

One challenging aspect of the VNCA is that all the parameters of the decoder are in the update function, which is iteratively applied $M(K + 1)$ times ($K$ doublings and $M$ NCA steps per doubling). This means that adding more parameters, and thus more computation, to the decoder is $M(K + 1)$ times more computationally costly than in a standard non-iterative decoder. In other words, we could have a decoder for CelebA with $M(K + 1) = 48$ times more parameters at the same computational cost. For this reason, the update function we use has few parameters, just 1.2M, compared to state-of-the-art decoders which routinely have hundreds of millions, allowing much more powerful decoders (Maaløe et al., 2019; Salimans et al., 2017; Vahdat & Kautz, 2020).

Another set of generative models that generates and reconstructs data in an iterative process are diffusion models (Sohl-Dickstein et al., 2015), which have recently achieved state-of-the-art results in datasets like CelebA (Ho et al., 2020; Kim et al., 2021). While the VNCA decoding processes involves applying a function repeatedly, the decoding processes in a diffusion models involves iteratively sampling from a Markov chain of approximate distribution. In the future, the connection to diffusion models should be explored in more detail.

## ACKNOWLEDGEMENTS

This work was funded by a DFF-Research Project1 grant (9131- 00042B), the Danish Ministry of Education and Science, Digital Pilot Hub and Skylab Digital.

## REPRODUCIBILITY STATEMENT

The code to reproduce every experiment is available at github.com/rasmusbergpalm/vnca. The code used to run all experiment, including all hyper-parameters, are recorded in the immutable git revision history. The revisions corresponding to the experimental results in the paper are highlighted in the repository README.

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

# A APPENDIX

## A.1 DAMAGE DURING GROWTH

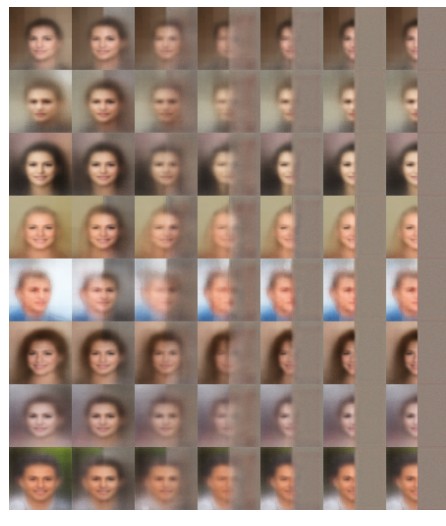 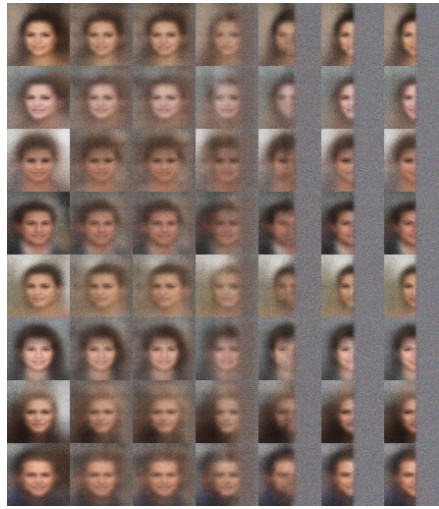

(a) VNCA                                  (b) Deep convolutional decoder baseline

Figure 11: Unconditional samples, after being exposed to damage at varying stages of growth. The leftmost image is not exposed to any damage. The next six images represent damage applied at steps $t = [0, 8, 16, 24, 32, 40]$ corresponding to the initial state, and the steps just before each doubling. (b) shows the equivalent process in a deep convolutional baseline: the leftmost image is left intact, and the following columns present the result of damaging the hidden state before each up-convolution. The second and third columns are the same, since both represent the equivalent to damaging the state before the first duplication. Note: both these figures show sample averages for clarity.

**CelebA baseline** To compare against resilience to damage in the decoding/doubling process (see Fig. 11), we trained a deep convolutional VAE on the same dataset, fitting the same distribution. This is the specification of the model as a PyTorch module:

```
VAE(
  (encoder): Sequential(
    (0): Conv2d(3, 32, kernel_size=(5, 5), stride=(1, 1), padding=(2, 2))
    (1): ELU(alpha=1.0)
    (2): Conv2d(32, 64, kernel_size=(5, 5), stride=(2, 2), padding=(2, 2))
    (3): ELU(alpha=1.0)
    (4): Conv2d(64, 128, kernel_size=(5, 5), stride=(2, 2), padding=(2, 2))
    (5): ELU(alpha=1.0)
    (6): Conv2d(128, 256, kernel_size=(5, 5), stride=(2, 2), padding=(2, 2))
    (7): ELU(alpha=1.0)
    (8): Conv2d(256, 512, kernel_size=(5, 5), stride=(2, 2), padding=(2, 2))
    (9): ELU(alpha=1.0)
    (10): Flatten(start_dim=1, end_dim=-1)
    (11): Linear(in_features=8192, out_features=512, bias=True)
  )
  (decoder_linear): Sequential(
    (0): Linear(in_features=256, out_features=4096, bias=True)
    (1): ELU(alpha=1.0)
  )
  (decoder): Sequential(
    (0): ConvTranspose2d(1024, 512, kernel_size=(5, 5), stride=(2, 2),
            padding=(2, 2), output_padding=(1, 1))
    (1): ELU(alpha=1.0)
    (2): ConvTranspose2d(512, 256, kernel_size=(5, 5), stride=(2, 2),
            padding=(2, 2), output_padding=(1, 1))
```

```
    (3): ELU(alpha=1.0)
    (4): ConvTranspose2d(256, 128, kernel_size=(5, 5), stride=(2, 2),
            padding=(2, 2), output_padding=(1, 1))
    (5): ELU(alpha=1.0)
    (6): ConvTranspose2d(128, 64, kernel_size=(5, 5), stride=(2, 2),
            padding=(2, 2), output_padding=(1, 1))
    (7): ELU(alpha=1.0)
    (8): ConvTranspose2d(64, 32, kernel_size=(5, 5), stride=(2, 2),
            padding=(2, 2), output_padding=(1, 1))
    (9): ELU(alpha=1.0)
    (10): ConvTranspose2d(32, 10, kernel_size=(5, 5), stride=(1, 1),
            padding=(2, 2))
  )
)
```

During training, we used the ELBO loss with $\beta = 100$ since we faced similar issues when trying to get sensible unconditioned samples. It is worth remarking that this baseline doubles the image via up-convolutions the same number of times as the VNCA used for this experiment.

## A.2   NON-DOUBLING VNCA VARIANT

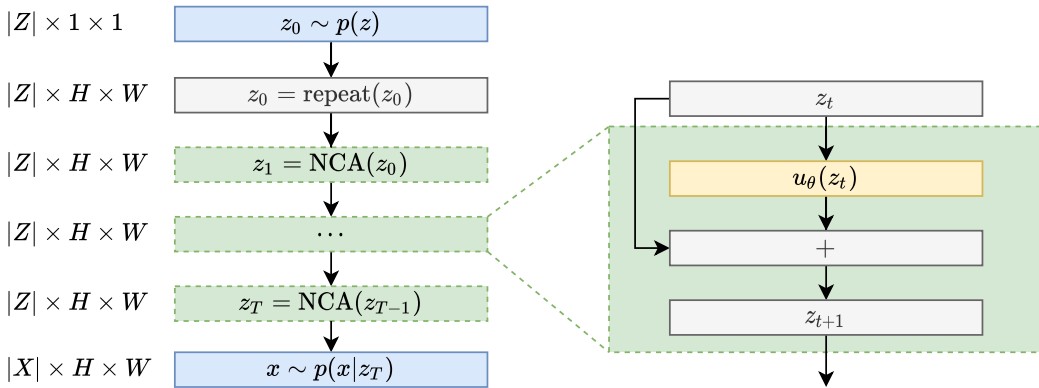

Figure 12: Non-doubling VNCA variant architecture.

## A.3 POOL AND DAMAGE TRAINING

```
1  import random
2
3  def pool_dmg_train(p_z_0, data, batch_size, pool_size, optim):
4      pool = []
5      while not converged():
6          x = random.sample(data, batch_size)
7          n_pool_samples = batch_size // 2
8          z_pool = None
9          if len(pool) > n_pool_samples: # Sample from the pool
10             x_pool, z_pool = pool[:n_pool_samples]
11             x[n_pool_samples:] = x_pool
12             z_pool = damage_half(z_pool)
13         q_z_0, z_0 = encode(x)
14         if z_pool:
15             z_0[n_pool_samples:] = z_pool
16         p_x_given_z_T, z_T = NCA(z_0) # Run the NCA
17         L = ELBO(p_x_given_z_T, x, q_z_0, p_z_0)
18         L.backwards()
19         optim.step()
20         # Add new states to pool
21         pool.append((x, z_T))
22         random.shuffle(pool)
23         pool = pool[:pool_size]
```

### A.4 NOTO EMOJI

The Noto Emoji font contains 2656 vector graphic emojis[2]. This dataset allows for comparison with previous NCA auto-encoder work, which uses it (Frans, 2021; Chen & Wang, 2020; Ruiz et al., 2020; Mordvintsev et al., 2020). Also, the emoji are diverse, which tests the ability of the VNCA to generate a large variety of outputs encoded in a single latent vector format. However, due to the small amount of samples and the very large variation, it is not a great dataset for evaluating a generative model. We train a VNCA on $64 \times 64$ emoji examples using $K = 5$ doublings and using 20% (531) of the images as a test set.

The VNCA learns to reconstruct the inputs well — similarly to the auto-encoder NCAs (Frans, 2021; Chen & Wang, 2020; Ruiz et al., 2020) — as can be seen on the left but fails to generate realistic-looking samples, as can be seen on the right

(Fig. 13). It does particularly well on the faces and human emojis, probably because they make up a large part of the dataset and are the most regular. While it reconstructs most images well, it fails on some, in particular the signs and symbols that are unique, not resembling any other emojis in the dataset, e.g., a highway, a safety pin, and a microscope. While the VNCA here fails to learn a good generative model, we are impressed with its ability to generate such diverse outputs from a single common vector encoding, using a simple self-organising process.

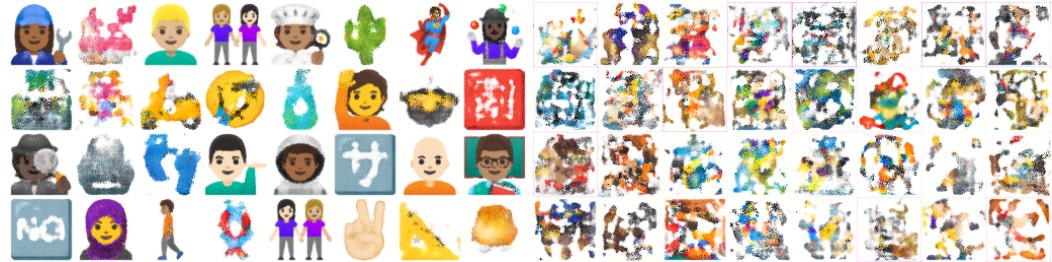

Figure 13: Noto Emoji results. Left: Test set reconstructions. Right: Unconditional samples from the prior. The VNCA achieves $3.09$ bits per dimension on the test set evaluated with 128 importance weighted samples.

---

[2]https://github.com/googlefonts/noto-emoji

## A.5 MODEL SPECIFICATION

The detailed model specification can be seen below in the standard PyTorch Module format.

```
VNCA(
  (encoder): Sequential(
    (0): Conv2d(3, 32, kernel_size=(5, 5), stride=(1, 1), padding=(2, 2))
    (1): ELU(alpha=1.0)
    (2): Conv2d(32, 64, kernel_size=(5, 5), stride=(2, 2), padding=(2, 2))
    (3): ELU(alpha=1.0)
    (4): Conv2d(64, 128, kernel_size=(5, 5), stride=(2, 2), padding=(2, 2))
    (5): ELU(alpha=1.0)
    (6): Conv2d(128, 256, kernel_size=(5, 5), stride=(2, 2), padding=(2, 2))
    (7): ELU(alpha=1.0)
    (8): Conv2d(256, 512, kernel_size=(5, 5), stride=(2, 2), padding=(2, 2))
    (9): ELU(alpha=1.0)
    (10): Flatten(start_dim=1, end_dim=-1)
    (11): Linear(in_features=8192, out_features=512, bias=True)
  )
  (nca): DataParallel(
    (module): NCA(
      (update_net): Sequential(
        (0): Conv2d(256, 256, kernel_size=(3, 3), stride=(1, 1), padding=(1, 1))
        (1): Residual(
          (delegate): Sequential(
            (0): Conv2d(256, 256, kernel_size=(1, 1), stride=(1, 1))
            (1): ELU(alpha=1.0)
            (2): Conv2d(256, 256, kernel_size=(1, 1), stride=(1, 1))
          )
        )
        (2): Residual(
          (delegate): Sequential(
            (0): Conv2d(256, 256, kernel_size=(1, 1), stride=(1, 1))
            (1): ELU(alpha=1.0)
            (2): Conv2d(256, 256, kernel_size=(1, 1), stride=(1, 1))
          )
        )
        (3): Residual(
          (delegate): Sequential(
            (0): Conv2d(256, 256, kernel_size=(1, 1), stride=(1, 1))
            (1): ELU(alpha=1.0)
            (2): Conv2d(256, 256, kernel_size=(1, 1), stride=(1, 1))
          )
        )
        (4): Residual(
          (delegate): Sequential(
            (0): Conv2d(256, 256, kernel_size=(1, 1), stride=(1, 1))
            (1): ELU(alpha=1.0)
            (2): Conv2d(256, 256, kernel_size=(1, 1), stride=(1, 1))
          )
        )
        (5): Conv2d(256, 256, kernel_size=(1, 1), stride=(1, 1))
      )
    )
  )
)
```

## A.6 DEEP CONVOLUTIONAL BASELINES

**MNIST baseline** In Fig. 5 we compared our VNCA to a deep convolutional VAE baseline. Here is the specification of that model using PyTorch's module format:

```
VAE(
  (encoder): Sequential(
```

```
  (0): Conv2d(1, 32, kernel_size=(5, 5), stride=(1, 1), padding=(4, 4))
  (1): ELU(alpha=1.0)
  (2): Conv2d(32, 64, kernel_size=(5, 5), stride=(2, 2), padding=(2, 2))
  (3): ELU(alpha=1.0)
  (4): Conv2d(64, 128, kernel_size=(5, 5), stride=(2, 2), padding=(2, 2))
  (5): ELU(alpha=1.0)
  (6): Conv2d(128, 256, kernel_size=(5, 5), stride=(2, 2), padding=(2, 2))
  (7): ELU(alpha=1.0)
  (8): Conv2d(256, 512, kernel_size=(5, 5), stride=(2, 2), padding=(2, 2))
  (9): ELU(alpha=1.0)
  (10): Flatten(start_dim=1, end_dim=-1)
  (11): Linear(in_features=2048, out_features=256, bias=True)
)
(decoder_linear): Sequential(
  (0): Linear(in_features=128, out_features=2048, bias=True)
)
(decoder): Sequential(
  (0): ConvTranspose2d(512, 256, kernel_size=(5, 5), stride=(2, 2),
        padding=(2, 2), output_padding=(1, 1))
  (1): ELU(alpha=1.0)
  (2): ConvTranspose2d(256, 128, kernel_size=(5, 5), stride=(2, 2),
        padding=(2, 2), output_padding=(1, 1))
  (3): ELU(alpha=1.0)
  (4): ConvTranspose2d(128, 64, kernel_size=(5, 5), stride=(2, 2),
        padding=(2, 2), output_padding=(1, 1))
  (5): ELU(alpha=1.0)
  (6): ConvTranspose2d(64, 32, kernel_size=(5, 5), stride=(2, 2),
        padding=(2, 2), output_padding=(1, 1))
  (7): ELU(alpha=1.0)
  (8): ConvTranspose2d(32, 1, kernel_size=(5, 5), stride=(1, 1), padding=(4, 4))
)
)
```

We believe this is a fair baseline, since it contains 9.5M trainable parameters, in comparison to the VNCA used for this experiment, which contains around 5M.

## A.7 CELEBA UNCONDITIONAL SAMPLES

## A.8 ADDITIONAL EXPERIMENTS

We tried several variations of the VNCA that, perhaps surprisingly, were not better. We report them here so that future research can take them into account.

**Size** We tried deeper (up to 8 residual layers) and wider (512) update function networks, with no effect. A bigger latent space of 512 didn't seem to matter. A bigger NCA hidden state of 512 also didn't seem to matter. We tried $L = \{1, 4\}$ logistic mixtures, and $L = 1$ was better.

**Non-strict NCA architectures.** We tried replacing the $1 \times 1$ convolution layers with $3 \times 3$, which means the NCA is no longer relying only on local communication, but this also wasn't much better. We also tried a version that had a separate update network per scale, i.e., $K$ networks. Surprisingly this was only slightly better.

**Batch size** Batch size did, surprisingly, seem to matter. Bigger batches consistently trained a lot better. With batch size 4 it was impossible to train. Batch size 32 seemed sufficient and was chosen as a trade-off between speed and quality.

**Update function**: We tried 1) A GRU update function, 2) $z_{t+1} = z_t + \sigma(b)u(z_t)$, where $b$ was a learned scalar and $\sigma$ is the sigmoid and 3) a version which held half of the cell hidden state constant throughout all updates. None were better.

## A.9 CELEBA RECOVERY

## A.10 NON-DOUBLING GROWTH

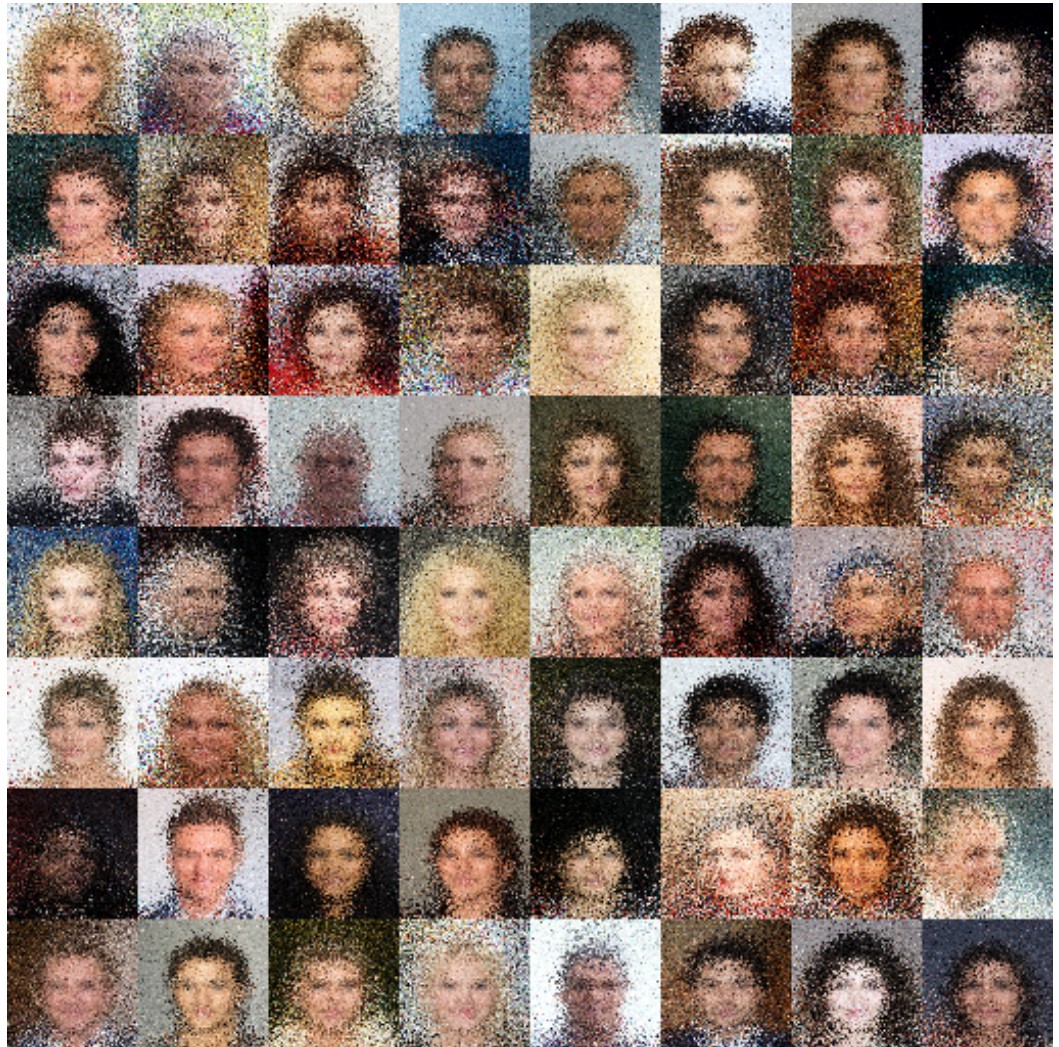

Figure 14: CelebA unconditional individual samples from a VNCA trained with $\beta = 100$.

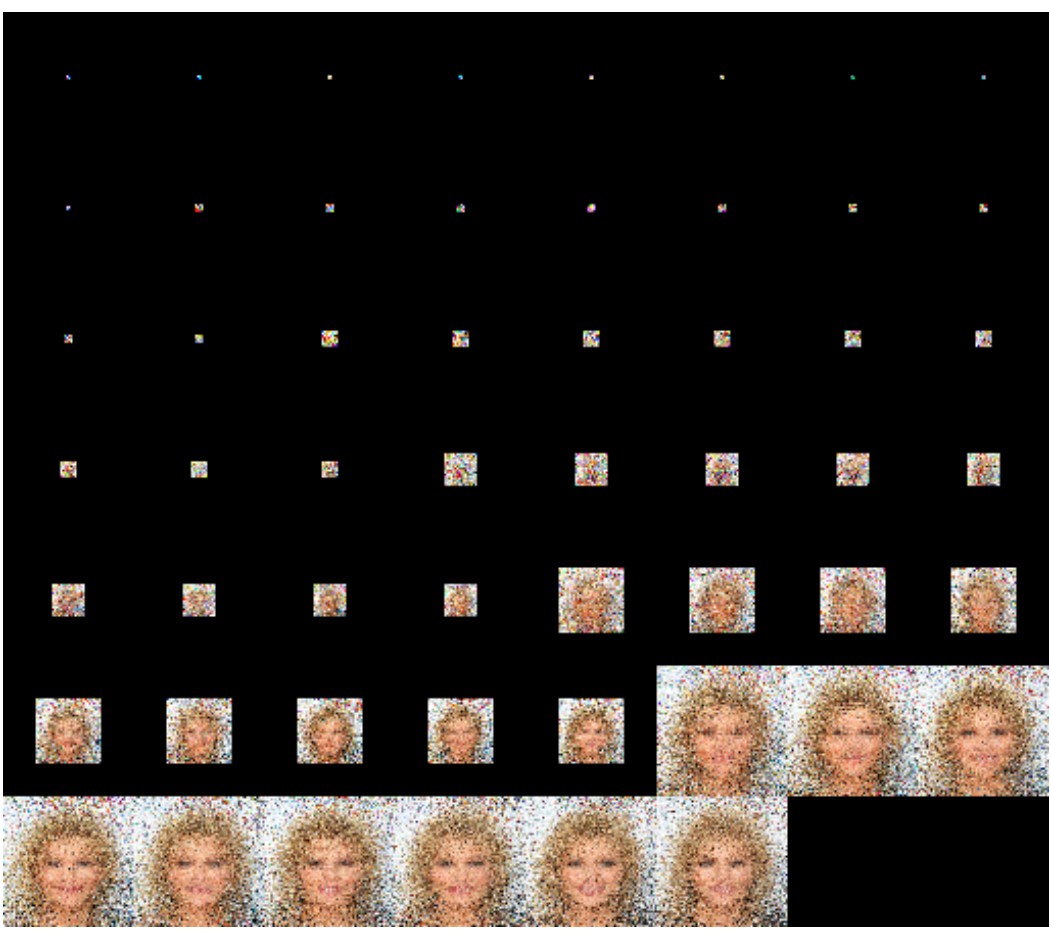

Figure 15: CelebA growth of unconditional sample from a VNCA trained with $\beta = 100$.

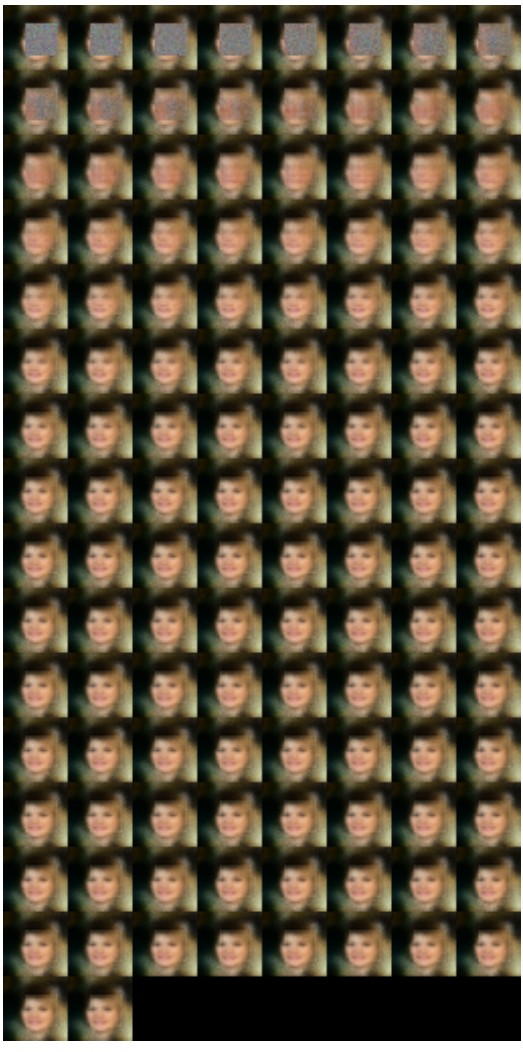

Figure 16: CelebA recovery of damage showing $T = 121$ NCA steps. Note: this shows sample averages for clarity.

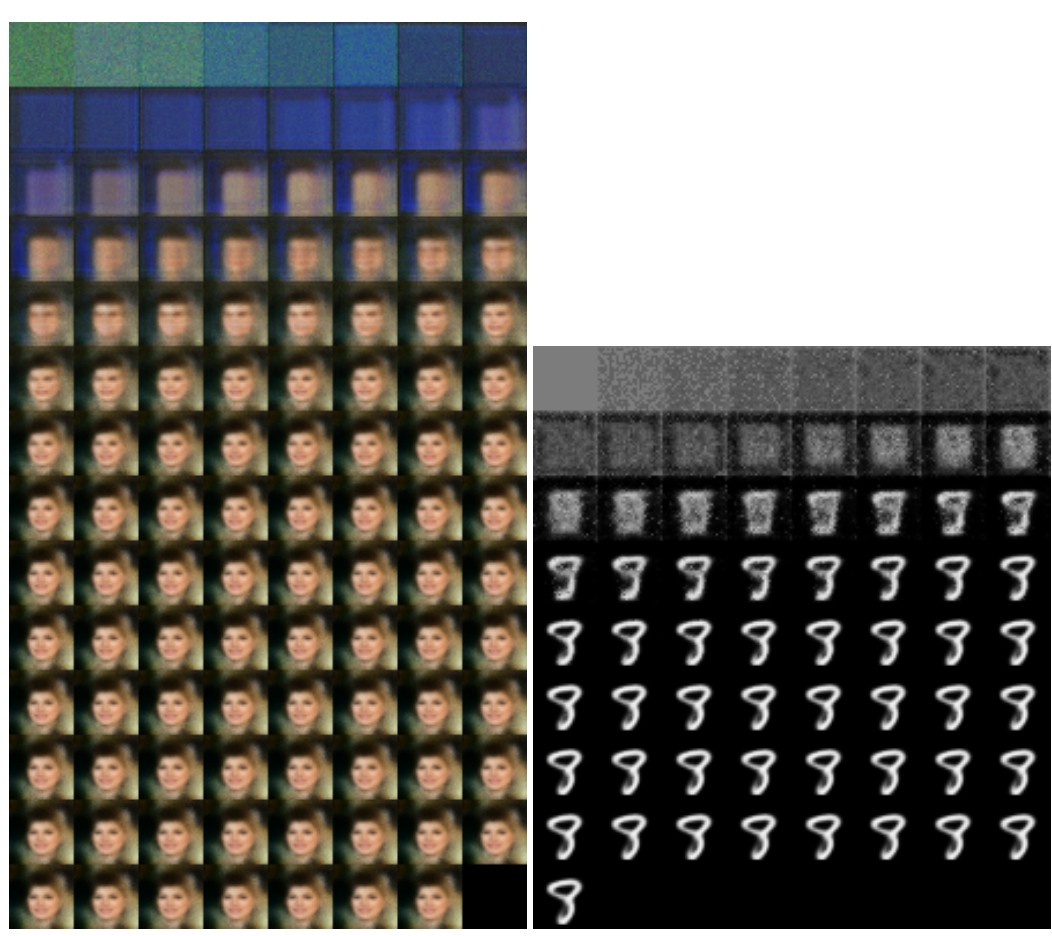

Figure 17: Growth of CelebA and MNIST digits for the non-doubling VNCA variant.

