# OpenReview forum: "Variational Neural Cellular Automata"
_ICLR.cc/2022/Conference — ICLR 2022 Poster_

### Official Review · Reviewer_cp9d · 2021-10-20

**Correctness:** 3
**Technical Novelty And Significance:** 3
**Empirical Novelty And Significance:** 2
**Recommendation:** 5
**Confidence:** 4

**Main Review:**

Pros:

- Figure 2 is extremely clear, and I understand pretty much how the model works.

- I appreciate that the authors looked at a number of different datasets to compare their model against.

- I think the "damage" experiments in Figure 10 were very interesting and well done, and could have been more of a showcase of the paper.

Cons:

- I think this work could be improved by showing more tasks this approach would work very well for. For example, in Figure 10, what does neuron/latent variable ablation look like with a normal variational autoencoder with upsampling? Can this be quantified with log-likelihood? This would be a great metric to compare against.

- For the decoding process figures (Figure 1 and 4), I don't see how this approach necessarily differs from a regular variational autoencoder with some sort of upsampling.

- With respect to inpainting of images, there are many other approaches that directly use the log-likelihood that could be compared against.

- I don't think the claims of "super-resolution" are well founded, especially if those claims are only using MNIST as the input dataset and the reconstructions have artifacts. This could be sufficiently tested by training the model either blurry or downsampled images and comparing the "super-resolved" images to the true high-resolution images.

Neutral:

- The individual samples reported in the appendix are a bit surprising (their low quality). I think this should have been reported more in the main part of the paper.


**Summary Of The Paper:**

The authors incorporate Neural Cellular Automata (NCA) into the decoding process of a variaitional autoencoder. They then train this model on a number of toy datasets, test out beta-VAE to make better reconstructions, and show a few reconstruction/decoding tasks that highlight the contributions of the described approach.

**Summary Of The Review:**

Generally, I think this model is an incremental improvement over neural cellular automata. While I do think that this approach is interesting and the results are sound, I think there could be more careful analysis of where the VNCA shines compared to existing approaches.

---

> ### Author Response · Authors · 2021-11-22
> **Rebuttal**
>
> First of all, thank you for the review and the kind comments regarding writing and clarity. We have now made it more clear where the VNCA shines compared to other approaches (see below for details).
>
> Following the reviewer’s suggestion, we have now significantly expanded the damage experiments. Interestingly, we found that both a deconvolutional VAE trained on CelebA and the VNCA have similar damage resilience during growth/up-convolutions. For the VNCA we found that specifically training for damage recovery allows it to be very resilient to damage and learn stable attractors around the data (see newly added Section 3.3. and Fig. 9).
>
> We’ve also added a latent space analysis and compared it with another similar baseline, which shows improved interpolations and clustering in latent space. Fig. 5 illustrates this comparison.
>
> The VNCA differs from a regular variational autoencoder with some sort of upsampling, in the sense that it’s the same function that is applied iteratively. In a regular VAE with upsampling each layer implements a different function, whereas in the VNCA it’s the same function that is repeatedly applied. Additionally, each step of VNCA update only relies on the local 3x3 neighborhood. The VNCA must learn a simple function that only relies on this local communication through many iterations.
>
> With regards to inpainting, we cannot directly compare to such methods, and have added a small explanation in the new Section 3.3 of why this is the case, repeated here: “While this looks like in-painting or data-imputation, it is not, and we should be careful not to compare to such methods. When doing in-painting the method only has access to the pixels of the damaged image, whereas here, each pixel corresponds to a vector valued cell state, that can potentially encode the contents of the entire image. Indeed this seems to be the case, since the VNCA almost perfectly recover the original input.”
>
> With regards to super-resolution, we agree and have removed the claims entirely. The experiments did not support them. The suggestions by the reviewer sounds like a good way of testing super-resolution, which we believe is interesting future work. Also, see our comment regarding super-resolution to reviewer kwgv.
>
> We tried to be clear in regards to when we were showing actual samples from p(x|z), and when we were showing averages of p(x|z). We have added a small paragraph to make the distinction between samples and averages more explicit. Also, we’ve added a note on every figure when we’re showing means. Note that the very noisy celebA samples in the appendix is for the VNCA trained with beta=100, which is heavily weighted to focus on low KL loss, compared to clear reconstructions. The CelebA reconstructions and samples from a VNCA trained with beta=1 in figure 7 are much less noisy.

---

> > ### Author Response · Authors · 2021-11-28
> > **Reminder**
> >
> > Dear Reviewer. Have you had a chance to look at our rebuttal and updated paper? We're eagerly awaiting your response.

---

### Official Review · Reviewer_AnwX · 2021-10-20

**Correctness:** 4
**Technical Novelty And Significance:** 2
**Empirical Novelty And Significance:** 1
**Recommendation:** 5
**Confidence:** 4

**Main Review:**

As the authors rightly point out, cellular automata and self-organizing systems are interesting directions of research. As I explain below, the main insight that can be derived from this paper (although not explicitly mentioned) is that a class of deep convolutional networks (networks that have repeated blocks and satisfy some other requirements) can be interpreted as neural cellular automata. With this alternative perspective, much of what the paper shows is not surprising and the questions that do arise are not addressed.

Here are a number of questions that the paper does not address:

- How does the decoder architecture proposed here differ from other decoder architectures proposed in the past?
The full architecture of the decoder network (including all the NCA steps) can be rolled into a single convolutional network with upsampling. From this perspective, the decoder architecture has repeating blocks (which allows it to be interpreted as an NCA). As a deep convolutional network, it is no surprise that the proposed decoder can function as a part of a VAE. Furthermore, I believe that many deep convolutional networks with repeating blocks (say deep resents) can also be interpreted as NCAs. Therefore, what seems to be novel in this paper (in the eyes of this reviewer) is the NCA *interpretation* and not the architecture itself.

- What does one gain by the NCA interpretation of such deep conv net?
As mentioned above, the NCA decoder can equivalently be viewed as a deep net of a specific shape. From that perspective, the analyses of the paper are not novel. An important question for the paper is then: what does one gain by this NCA interpretation? How does this NCA type structure change the behavior of the conv net if at all?

- Is there a way to train this network while remaining true to the NCA principles?
The way the entire system is trained is using backprop through time. This goes against the ideals of self-organizing systems to some extent. Is there any way to train this system such that the updates to the weights of each automaton are also local?

**Summary Of The Paper:**

The submission proposes a VAE whose decoder is implemented via neural cellular automata (NCA). The authors show that this construction performs well when looking at the likelihood of the evidence (given samples) and show that the architecture has some robustness properties against damage during generation.

**Summary Of The Review:**

Unfortunately, almost all the technical and experimental demonstrations of the paper can be thought of as rather straight-forward consequences of the fact that the decoder is in the end a deep convolutional network. With this perspective, there is not a lot of novelty in the paper and the consequences of the fact that this deep convolutional network can in fact be implemented as cellular automata are not discussed.

---

> ### Author Response · Authors · 2021-11-22
> **Rebuttal**
>
> We are happy that the reviewer agrees that NCAs are an interesting research direction.
>
> However, we disagree with the reviewer that the main insight of our paper is that a class of deep convolutional networks can be interpreted as neural cellular automata; this insight was already well known and was referenced in our paper (Gilpin 2019, Mordvintsev et al. 2020). For instance, Mordvintsev et. al., writes: “The close relation between Convolutional Neural Networks and Cellular Automata has already been observed by a number of researchers [16, 17]. The connection is so strong it allowed us to build Neural CA models using components readily available in popular ML frameworks. Thus, using a different jargon, our Neural CA could potentially be named “Recurrent Residual Convolutional Networks with ‘per-pixel’ Dropout”.
>
> The main insight introduced in our paper is that **NCAs can model complex data distributions through morphogenesis**, and show resilience to damage **despite** their computational constraints. Showing that NCAs can model complex data distributions is to the best of our knowledge novel.
>
> The reviewer is right that the entire VNCA can be unrolled into a very deep convolutional neural network, with some upsampling and a lot of weight sharing. In the same sense that a Recurrent Neural Network can also be seen as a very deep MLP with weight sharing.
> The reviewer is not correct, however, that e.g. resnets can also be seen as an NCA since the functions of the residual layers are not repeated. A deep residual network does not learn a single function that is iterated for many time steps, rather it learns a lot of functions that when composed produce the desired output. In short deep resnets learn a function of the form y=f(g(h(...j(x)))), and the VNCA learns a function of the form y=f(f(f(...f(x)))).
>
> We have added a new Section 3.3 that examines the ability of the VNCA to recover from damage, by exactly using it’s ability to keep iterating the same function over and over. This clearly shows that the VNCA can do something a standard deep resetnet decoder cannot, which is to keep growing, and recovering from damage after growth.
>
> With regards to training the VNCA in a local-only manner, we agree that this is an interesting direction for future research. However, we do not agree that the current setup goes against the ideas of a self-organizing system. No current NCA setup that we are aware of is trained in a local-only manner and the same holds true for the biological inspiration for NCAs: biological organisms are grown through the local interaction of cells but the genome that guides this process has not been trained local-only but through evolution. In summary, while the generative process is self-organizing and through local interactions (in biological and artificial CAs), the training process is not. We’ll make this distinction more clear in the paper.
>
> We sincerely hope the reviewer will reconsider their rating in light of the rebuttal and the new experiments.
>
> [16] - Wulff, N., and J. A. Hertz. "Learning cellular automaton dynamics with neural networks." Advances in Neural Information Processing Systems 5 (1992): 631-638.
> [17] - Gilpin, William. "Cellular automata as convolutional neural networks." Physical Review E 100.3 (2019): 032402.

---

> > ### Comment · Reviewer_AnwX · 2021-11-23
> > **Response to initial rebuttal**
> >
> > I thank the authors for their rebuttal.
> >
> > I would like to point out to the authors that resnets with constant weights (i.e. weights tied across all layers) have been studied in the past, especially in the context of neural ODEs (see [1] as an example). Therefore I maintain that many (not all) of the claims in the paper are either explicitly known in literature in one form or another or are not surprising given what is out there.
> >
> > However, I think there is value in bridging the literature between what is known in the resnet/nODE domain and NCA. I am increasing my score but I would like to see the authors explicitly explain the relationship between NCA and weight shared resnets and give references to prior work.
> >
> > [1] Avelin, Benny, and Kaj Nyström. "Neural ODEs as the deep limit of ResNets with constant weights." Analysis and Applications 19, no. 03 (2021): 397-437.

---

> > > ### Author Response · Authors · 2021-11-24
> > > **Response**
> > >
> > > We thank the reviewer for their response and for updating their score.
> > >
> > > Thank you for the very interesting reference. We will happily add a small discussion of this connection to constant weight resnets/nODE in the camera-ready version.

---

### Official Review · Reviewer_kwgv · 2021-10-29

**Correctness:** 4
**Technical Novelty And Significance:** 3
**Empirical Novelty And Significance:** Not applicable
**Recommendation:** 8
**Confidence:** 5

**Main Review:**

**Strengths**

- I enjoyed the paper and, although I have some comments, I think the work can be interesting for the ICLR community. There is growing interest in (neural) cellular automata, and this work is a valuable addition to the topic.

- The paper addresses a limitation of typical morphogenesis approaches with NCA, which is that they are usually trained to generate a single image/state. This work is a first step towards learning NCA that generalize to different target states.

- The experimental results are interesting despite having some limitations. The VNCA can reconstruct the images well when starting from a latent sample produced by the encoder. The limitations only arise when using the model for generation starting from a random sample from the prior, but the authors also show how to partially address these issues (using the $\beta$-VAE approach).

- The paper is well written and all concepts are explained adequately.

**Weaknesses**

I have a few suggestions and questions that I hope can help the authors improve the paper.

1. Have the authors tested their model without the upsampling part?

    The base NCA of Mordvintsev et al. is a sufficiently powerful architecture for morphogenesis (starting from a single nonzero pixel) and it is not immediately clear why the upsampling step is necessary for the VNCA. An explanation of what led the authors to this choice (besides the link with mitosis) would be interesting.

2. As noted by the authors, the upsampling step forces the output to have a given dimension, multiple of $2^M$.

    Related to this, have the authors investigated some settings in which the images are exactly as big as the output of the VNCA, without zero padding? This would give us interesting insights into the behaviour of the VNCA at the boundaries.

3. I have a few questions regarding the architecture:

    - What is the activation function of the last layer of the NCA?
    The authors only mention ELU but the last activation should ensure that the output of the model is a meaningful image (so a sigmoid or rescaled tanh).
    - How was the NCA architecture chosen? Why four residual blocks? Why two 1x1 convolutions in each block?
    - Why is there no activation after the last convolution of each residual block?
    - How was the architecture of the encoder chosen?
    - How were the number of channels, learning rate, batch size, gradient clipping, $K$, $L$, and $\beta$ determined?
      I'm not asking the authors to perform an in-depth tuning, but if the hyperparameters were chosen arbitrarily it is important that the readers know it.
    - How do 1.2M parameters compare to Mordvintsev at al.'s NCA?

5. In Figure 4, the final generated digits are significantly smoother than those in Figure 3, right panel. Were the examples of Figure 4 cherry-picked somehow?

6. There are no details about the training procedure. For example, Mordvintsev et al. use a replay memory to ensure the stability of the generated pattern. Do the authors have a comparable training procedure in place?

    NCA are not well known in the community (yet), so papers on the subject must a good job of explaining the base concepts until they are more widely adopted.

7. Two references could/should be added in Section 1:

    - Wulff and Hertz, "Learning cellular automaton dynamics with neural networks", Neural Information Processing Systems (1992).
       This is one of the first papers to model CA transition rules with neural networks (they even use CNNs!) and a good reference to cite for completeness.

    - Grattarola et al., "Learning Graph Cellular Automata", Neural Information Processing Systems (2021).
       This is a very recent paper (published after the ICLR submission date) that explores the use of graph neural networks to learn cellular automata on graphs (they also do morphogenesis). It seems like a good match for the last paragraph of Page 2, which has no references.

**Other comments**

- Some readers may not be familiar with the "bits per pixel" metric. An explanation of what the metric represents could be useful.

- Regarding this claim: "It is not immediately clear what a generative model that takes damage and super-resolution into account would look like".
I don't think that NCA are that different from any typical neural network. Couldn't we train the VNCA as any other super-resolution or de-noising autoencoder?

    For example, the de-noising could be done as they do in the NCA paper, and the super-resolution by fixing two different generations at which the loss is computed, using the target image and a downscaled version of it.

- Have the authors performed a latent space analysis to see if different classes of images get clustered in some particular way?

- There is no mention of code anywhere in the submission. I encourage the authors to make the code available if the paper gets accepted (ideally it should have been included with the submission, but I understand that it is not mandatory and I have not taken this into consideration when evaluating the paper).

----------

**After rebuttal**: The authors have addressed my comments. I have raised my score from "6: marginally above the acceptance threshold" to "8: accept, good paper".

**Summary Of The Paper:**

This paper introduces a variational generative model based on Neural Cellular Automata (NCA). The model is called the Variational NCA (VNCA).

The VNCA is designed for images:

- The encoder is a typical convolutional neural network.
- The decoder is an NCA that iteratively refines the image, alternating convolutions and upsampling up to the desired image size. The upsampling process is loosely inspired by mitosis in living cells.

The authors perform morphogenesis experiments on three datasets: MNIST, Noto Emoji, and CelebA.
The results are good on MNIST, less so on the other two datasets, although there is clear evidence that the model can learn to generate meaningful images.

The authors also perform an experiment to see if the VNCA is robust to perturbations (occlusions) and show that the model has a reasonable degree of robustness even without ever seeing any perturbations at training time.

**Summary Of The Review:**

- The authors propose a variational neural cellular automaton, which learns to generate images by iterating the transition rule.
- The paper is interesting, with good results, and a good fit for ICLR.
- The paper solves an interesting problem on the topic of neural cellular automata.
- There are some doubts/limitations that I have asked the authors to address (mainly concerning the architecture of the model).
- There are some missing references and details that would help the readers to get a better sense of the subject.

-----
**After rebuttal**: The authors have addressed my comments. I have raised my score from "6: marginally above the acceptance threshold" to "8: accept, good paper".

---

> ### Author Response · Authors · 2021-11-22
> **Rebuttal part 1**
>
> We thank the reviewer for the thorough review. We are happy the reviewer finds the paper interesting and valuable to the community. We have now added additional experiments that we believe further improved the paper, in particular a variant that is robust to damage and learns stable attractors around the data. We have also added an analysis of the latent space, as suggested by the reviewer.
>
> Weaknesses
>
>  - We have now added a new section to the paper (Section 3.3), which explores a non-doubling VNCA for damage recovery. In short, it’s much better suited for damage recovery and long-term stability but it is a worse generative model. We hypothesize that (1) the doubling operation is a good inductive bias on image data, and that (2) it allows the cells to agree on a global structure early on. The communication in the early growth steps effectively enables the doubling VNCA to communicate over, what becomes, large distances, while the non-doubling NCA would need many steps to achieve something similar. The doubling VNCA also has the benefit that it is computationally cheaper since it’s only in the last stage of growth that the computations are performed on the entire image. Compared to a regular non-doubling NCA this results in a large speedup and lowers the memory requirements dramatically. We have added a paragraph on this in the discussion. Thanks!
>  - Yes, the output will be of 2^M, but it’s easy to model any data size H x W, by growing to 2^M > H x W and then cropping to H x W. Alternatively the data can be padded to the nearest 2^M, which is the approach we took for MNIST (which is 28x28).
>  - Yes. If we understand the question correctly, this is the case for CelebA which is 64 x 64, so exactly 5 doublings from a 2x2 seed. It does not seem to affect the VNCA.
>
> Clarifications on the neural architecture:
>
>  - The last layer of the update network is linear, which ensures that the additive updates behave nicely and have good gradients. After the NCA has run the final NCA hidden states z_T condition a likelihood, as the reviewer notes. For MNIST we use the Binomial likelihood and the first channel of the NCA hidden state is used as the logits. In effect, this means we use a sigmoid. For the RGB datasets we use the Discretized Mixture of Logistic first introduced in PixelCNN++ (https://arxiv.org/pdf/1701.05517.pdf), which is a little complex. It uses the first 10*L channels of the NCA to determine the p(x|z) for each color channel, where L is the number of mixture distributions. However, we just use a single mixture so, it’s actually just a Discretized Logistic Distribution. The logistic distribution is the distribution which has a logistic function as the CDF. It’s discretized by summing the probability mass in 1/256 increments (assuming uint8 RGB). It takes two parameters for each channel: the mean and a scale, similar to a standard deviation. The 10 channels used corresponds to base means (3,) and scales (3,) for the RGB channels, plus a (softmax logit) mixture weight (1,) (which we don’t use, since we only have a single mixture). The last 3 channels corresponds to relative mean offsets coefficients, such that mean_R = base_mean_R, mean_G = base_mean_G + c_1 * mean_R, and mean_B = base_mean_B + c_2 * mean_R + c_3 * mean_G. For details see the PixelCNN++ paper.
>  - In general, the architectures and hyper-parameters were chosen by what worked best on the validation set, and fit on a single GPU. We’ve added this to the paper. Thanks. We tried many variations. See the appendix for some additional VNCA variants we investigated. The two 1x1 convolutions separated by a single non-linearity for each residual block were chosen since this is the minimal residual building block as proposed by the deep residual networks paper (https://arxiv.org/pdf/1512.03385.pdf).
>  - This is the way it’s usually done with residual networks, and also what was proposed in the resnet paper. If we used an ELU non-linearity after the last conv layer in the residual block, the residual block could only add positive numbers to the skipped data. If we used something like tanh, we could only change it by 1 in either direction. Having the last layer be linear means we can change it as much as is needed, and is the standard approach when using residual blocks, as far as we know. We have not tried using a non-linearity after the last conv layer in our residual blocks.
>  - The architecture of the encoder was chosen as a “typical” encoder in a simple VAE setting. This is admittedly a very loose definition. We did experiment with a slightly simpler encoder (less channels), which gave slightly worse results, but we did not try to otherwise tune the encoder.

---

> > ### Author Response · Authors · 2021-11-22
> > **Part 2**
> >
> >  - All those hyperparameters were chosen based on their performance on the validation set, and what could fit in the GPU. Usually we tried doubling/halving the parameters. The gradient clipping was needed to stabilize training. We’ve added a small remark on hyper-parameters in the start of the experiments section.
> >  - Mordvintsev at al.'s NCA has 8,336 trainable parameters. It’s so much smaller because they use a much smaller number of NCA hidden states and use a sobel filter without any learnable parameters in place of our learned 3x3 conv.
> >  - When displaying the results of a generative model we show both samples from p(x|z), i.e. x \sim p(x|z) and the average of p(x|z), i.e. E_{x \sim p(x|z)} x. We’ve added a small note about this in the experiments section to make it more clear, and updated all figure captions, so that we note if we’re showing averages. Figure 3 shows samples and Figure 4 show averages. The reason we show averages is that during growth the pixels are still quite uncertain (ilght grey), which results in noisy samples, which would obscure the analysis of the actual growth taking place.
> >  - We do not use pool training or any other tricks not mentioned for the doubling VNCA. In the new Section 3.3 with non-doubling VNCA trained for damage and stability, we do use pool training and induce damage during training. The training details are mentioned at the start of Section 3 and we included a pseudo-code algorithm in the appendix. We hope this will make it easier for the community to get familiarised with NCAs. Additionally, we’ll make all of our code available upon publication (see the added Reproducibility Statement).
> >  - Thank you for these relevant references. We have added them!
> >
> > Additional comments
> >
> >  - We have added a small definition of bits per pixel. Thanks!
> >  - Regarding damage recovery and super resolution: with the added Section 3.3 we’ve outlined our attempt at a generative model that takes damage into account. The difficulty is that in a generative model one must define a process which results in a data sample. As we describe in the section, we have to truncate the gradient computation to achieve stability over the very large number of steps required to recover from damage. Since we’re truncating the gradient we’re no longer exactly optimizing the ELBO, so we’re no longer strictly optimizing our generative model. In the end, it works out fine, but in general the damage recovery and super-resolution goal is somewhat tangential goals to the strictly generative modelling task. In super resolution the task is to produce a high resolution image, *conditioned* on the low resolution image, i.e. maximize p(high res|low res). That is fundamentally a different task than maximizing an *unconditional* generative process p(data). That being said, it would be interesting to actually train a NCA as a super-resolution model along the lines the reviewer mentioned!
> >  - Thank you for the suggestion! We have added a latent space exploration to the paper. In it we look at how latent interpolations decode to digits, and how the latent codes cluster using t-SNE. See Fig. 5 for a comparison between this structure in both our VNCA model, and a deep convolutional baseline for MNIST. These t-SNE embeddings show more structure and better separation of digits for the VNCA than for the deep convolutional baseline.
> >  - We agree and will make the code publicly available (see the Reproducibility Statement).

---

> > > ### Comment · Reviewer_kwgv · 2021-11-23
> > > **Raised score**
> > >
> > > I thank the authors for their reply.
> > >
> > > The authors have improved the paper significantly after the reviews, and they have addressed all questions and comments that I raised.
> > >
> > > Question 2 in my review was an oversight on my part. Thanks for the reply.
> > > I also thank the authors for clarifying my doubts regarding the architecture. As long as everything is properly reported in the paper, I don't see any further issues.
> > >
> > > I was already in favour of accepting the paper. After the rebuttal, I have raised my score to a full "accept".

---

### Official Review · Reviewer_rDYM · 2021-11-01

**Correctness:** 4
**Technical Novelty And Significance:** 2
**Empirical Novelty And Significance:** 3
**Recommendation:** 5
**Confidence:** 4

**Main Review:**

Pros:

1. The paper is very well-written, I could clearly follow the ideas and the experimental setting (modulo a few minor points mentioned at the end of this review). It should be reproducible.
2. It is interesting to see how good NCAs are in terms of likelihoods. Even the knowledge that they are not so good is valuable for the community (although I would prefer to see a few independent attempts like this to draw such a conclusion).
3. This is a very honest paper, it reveals all the shortcomings upfront and doesn't hide anything between the lines.

##########################################################################

Cons:

1. The experimental results are underwhelming. There is little evidence that the method actually works, except on MNIST.
2. The motivation of this work is not fully clear to me. What problem of prior work is this paper solving? I see that it introduces additional architectural limitations, and that's probably why the quality is much lower than SOTA. But what could we potentially gain?

##########################################################################

Questions during rebuttal period:

Significantly improving experimental results would make me reconsider the rating - not sure if this is feasible during rebuttal though.

#########################################################################

Minor suggestions and typos:

(1) "and several methods have been proposed." - maybe "several classes of methods", generative modeling is quite a vast field.

(2) "The Variational Auto-Encoder (VAE) is a seminal probabilistic generative model" - I would suggest citing both Kingma & Welling and Rezende et al., both papers appeared roughly at the same time. Up to you though.

(3) Van Oord -> van den Oord

(4) convolution based -> convolution-based

(5) "It achieves log p(x) >= -84.23" - is it in bits or nats? why not report bpd?

(6) "state-of-the-art, which is around -78.6" - assuming bits, it's not sota. See here: https://paperswithcode.com/sota/image-generation-on-mnist

(7) "Note: this shows the likelihood probabilities instead of samples from the likelihood for clarity." - likelihood probabilities sounds strange, why not just "probabilities"?

(8) "This is far from state-of-the-art which is around 2 bits per dimension" - what are the dimensions? Looking at https://paperswithcode.com/sota/image-generation-on-celeba-64x64, I assume it's 64x64?

**Summary Of The Paper:**

This paper proposes variational extension of Neural Cellular Automata for image generation. It performs experiments on MNIST, Noto Emoji and CelebA. The likelihood results are shown to be significantly behind SOTA on CelebA and also behind on other datasets. The paper provides qualitative analysis of the self-organized generation process and shows robustness to early-stage perturbations of latents.

**Summary Of The Review:**

I am slightly leaning towards rejecting this paper. While the writing is high-quality and the analysis is interesting, the practical results are underwhelming, both in sample quality (see Figure 7) and in likelihoods.

---

> ### Author Response · Authors · 2021-11-22
> **Rebuttal**
>
> First of all, thank you for the thorough review and the kind words regarding readability and reproducibility. In order to ensure reproducibility, we will release all the code, including all the hyper-parameters to reproduce all the experiments. See the newly added Reproducibility Statement in the paper.
>
> We’re happy the reviewer finds it interesting to see the likelihoods and results, regardless of whether they are state of the art or not. Our aim is to provide a fair and unbiased evaluation of the VNCA as a generative model. We think that it's important to publish regardless of the results, to limit publication bias, inspire others, and limit wasted duplicate work. This is a first attempt at using NCAs in generative modeling and very likely not the last word. We agree with the reviewer that we need to see more papers before drawing any final conclusions, and with that in mind, it's important that all the findings are actually published, regardless of whether the findings are positive or negative/mixed, as long as they are well motivated, fairly evaluated and reproducible.
>
> Thank you for valuing our honesty. This was very much our intention so we are pleased that the reviewer acknowledge this.
>
> With regards to motivation, prior work has been in the auto-encoding regime and has shown good reconstructions. However, prior work has never proposed a proper generative model, nor evaluated them fairly against state-of-the-art generative methods. The VNCA is the first proper generative model using NCAs and we evaluate it as fairly as possible. NCAs have several interesting properties such as only relying on the local interaction between cells, showing interesting dynamics despite their simplicity, and serving as a model for many biological phenomena.
>
> While clearly working better on MNIST, we would like to push back on the reviewer’s criticism that the method only works for this dataset. On CelebA, the VNCA  learns to reconstruct well (with a surprisingly low amount of parameters, compared to state-of-the-art VAEs), and with beta=100, it also learns to generate faces (see Figure 7). Although noisy due to high variance on each pixel, the mean samples clearly look like faces.
>
> With regards to new experimental results, we’ve added a significant amount of new material to the paper, regarding damage recovery (see Section 3.3). These new experiments show that the VNCA is very well capable of recovering from significant damage to an already grown image and learns stable attractors around the data (see Fig. 9). Interestingly, we found that both a deconvolutional VAE trained on CelebA and the VNCA have similar damage resilience during growth/equivalent up-convolutions (see Fig. 11).
>
> We’ve also added a latent space analysis and compared it with another similar baseline, which shows improved interpolations and clustering in latent space.
>
> Minor points:
>  - 1-4: Thanks! We have incorporated these changes.
>  - 5-6: For historical reasons log likelihoods on binarized MNIST is usually reported in nats summed over all the dimensions, not as bits per dimension. See https://paperswithcode.com/sota/image-generation-on-binarized-mnist. We followed the convention here to ease comparison to other work. The bpd can be computed by dividing by -(28*28*ln(2)). We have clarified this in the paper. The top paper on paperswithcode reports -76.93 nats.  We will add this reference and update the number. Thanks!
>  - 7: This was just to clarify we were showing the average of p(x|z), but we agree it sounded weird. We’ve made clear we’re showing averages.
>  - 8: Yes, it’s 64x64. We’ve made sure this is more clear in the paper.

---

> > ### Comment · Reviewer_rDYM · 2021-11-28
> > **Reviewer's response**
> >
> > Thank you for your clarifications. Since neither better samples, not better likelihoods are added to the paper, I am keeping my score (5).
> >
> > I am still leaning slightly towards rejection, due to underwhelming experimental results. Resilience to damage is not so practically interesting, and the need to average samples so that they look reasonable is certainly a minus.

---

> > ### Author Response · Authors · 2021-11-28
> > **Reminder**
> >
> > Dear Reviewer. Have you had a chance to look at our rebuttal and updated paper? We're eagerly awaiting your response.

---

### Author Response · Authors · 2021-11-22
**Rebuttal Summary**

We thank all the reviewers for their constructive feedback and suggestions. Based on this feedback we have added a substantial amount of new results and analysis that we believe improved the quality of our paper. The main additions are summarized below:

 - We have now added a new section to the paper (Section 3.3), which explores a non-doubling VNCA optimized for damage recovery. This model is much better suited for damage recovery (see Figure 9) and long-term stability but it is a worse generative model. Analyzing the pros and cons of these two models points to some interesting future research challenges.
 - We have added a latent space analysis and compared it with a baseline deconvolutional VAE. The result show improved interpolations and clustering in latent space of our VNCA version.
 - We tried to clarify some of the misunderstandings of reviewer AnwX. Mainly, that the insight introduced in our paper is not that deep convolutional networks can be interpreted as neural cellular automata (which was already known) but that that NCAs can model complex data distributions through morphogenesis, and show resilience to damage despite their computational constraints. We also clarified that NCAs are not the same as resnets, since the functions of the residual layers are not repeated.
 - We have moved the experiments on damage during growth to the appendix, since we found, to our surprise, that a standard VAE decoder was similarly resistant to damage during growth/up-convolutions. We have noted this in the main paper, since we find it surprising, and put the details in the appendix.

---

### Decision · Program_Chairs · 2022-01-20

**Decision:**

Accept (Poster)

**Comment:**

Meta Review for Variational Neural Cellular Automata

This paper proposes a generative model, a VAE whose decoder is implemented via neural cellular automata (NCA). The authors show that this model performs well for reconstruction, but they also show that the architecture has some robustness properties against damage during generation.

Experiments were conducted on 3 datasets: MNIST, Noto Emoji, and CelebA, and while experimental results were great on MNIST, the method was less performant so on the other two datasets, although there is clear evidence that the model can learn to generate meaningful images. For the robustness experiments, the authors show that VNCA is robust to perturbations (occlusions) and show that the model has a reasonable degree of robustness even without ever seeing any perturbations at training time.

All authors agree that this model is an improvement over neural cellular automata, and that the approach is interesting and the results are sound (and even useful). Initially, there were concerns that NCA's were simply convolutional neural networks (the connection is already known, and not the point of the paper), and issues with comparison with baselines for damage reconstruction tasks, but these were addressed by the authors (which the reviewers have acknowledged, and have improved their scores). The authors have also responded to the concerns of reviewer cp9d, and due to the lack of response from cp9d, I assessed the authors' response myself and find that they do address the concerns (in particular, they removed claims of super-resolution, and improved the clarity of the work). With that in mind, the score of 5 is viewed as a score of 6 from my perspective (giving this work effectively an average score of 6).

After my assessment of the paper and reviews, I agree with reviewer kwgv, as they have summarized the work in their original review:
- The authors propose a variational neural cellular automaton, which learns to generate images by iterating the transition rule.
- The paper is interesting, with good results, and a good fit for ICLR.
- The paper solves an interesting problem on the topic of neural cellular automata.
- There are some doubts/limitations that I have asked the authors to address (mainly concerning the architecture of the model).
- There are some missing references and details that would help the readers to get a better sense of the subject.

Crucially, kwgv have acknowledged that the *authors have improved the paper significantly after the reviews, and they have addressed all questions and comments that [they] raised* (especially with regards to the last 2 points), and kwgv has subsequently championed the work with a score of 8. With the increased scores from kwgv and AnwX in mind, and also with what I view as an increased score of 6 from cp9d (in the lack of response from the reviewer, the authors have addressed the concerns in my judgement), my conclusion is that this is a nice work that bridges NCAs with generative models, and I think the work will be a useful addition to the growing literature in this space. I will recommend it for acceptance at ICLR 2022 as a poster.